# Four-Week Exoskeleton Gait Training on Balance and Mobility in Minimally Impaired Individuals with Multiple Sclerosis: A Pilot Study

**DOI:** 10.3390/bioengineering12080826

**Published:** 2025-07-30

**Authors:** Micaela Schmid, Stefania Sozzi, Bruna Maria Vittoria Guerra, Caterina Cavallo, Matteo Vandoni, Alessandro Marco De Nunzio, Stefano Ramat

**Affiliations:** 1Bioengineering Laboratory, Department of Electrical, Computer and Biomedical Engineering, University of Pavia, 27100 Pavia, Italy; stefania.sozzi@unipv.it (S.S.); brunamariavittoria.guerra@unipv.it (B.M.V.G.); stefano.ramat@unipv.it (S.R.); 2Department of Sport, LUNEX, 4671 Differdange, Luxembourg; caterina.cavallo01@universitadipavia.it (C.C.); adenunzio@lunex.lu (A.M.D.N.); 3Luxembourg Health and Sport Sciences Research Institute A.s.b.l., 4671 Differdange, Luxembourg; 4Laboratory of Adapted Motor Activity (LAMA), Department of Public Health, Experimental Medicine and Forensic Science, University of Pavia, 27100 Pavia, Italy; matteo.vandoni@unipv.it

**Keywords:** gait exoskeleton training, Multiple Sclerosis (MS), quiet standing balance, TUG test

## Abstract

Multiple Sclerosis (MS) is a chronic neurological disorder affecting the central nervous system that significantly impairs postural control and functional abilities. Robotic-assisted gait training mitigates this functional deterioration. This preliminary study aims to investigate the effects of a four-week gait training with the ExoAtlet II exoskeleton on static balance control and functional mobility in five individuals with MS (Expanded Disability Status Scale ≤ 2.5). Before and after the training, they were assessed in quiet standing under Eyes Open (EO) and Eyes Closed (EC) conditions and with the Timed Up and Go (TUG) test. Center of Pressure (CoP) Sway Area, Antero–Posterior (AP) and Medio–Lateral (ML) CoP displacement, Stay Time, and Total Instability Duration were computed. TUG test Total Duration, sit-to-stand, stand-to-sit, and linear walking phase duration were analyzed. To establish target reference values for rehabilitation advancement, the same evaluations were performed on a matched healthy cohort. After the training, an improvement in static balance with EO was observed towards HS values (reduced Sway Area, AP and ML CoP displacement, and Total Instability Duration and increased Stay Time). Enhancements under EC condition were less marked. TUG test performance improved, particularly in the stand-to-sit phase. These preliminary findings suggest functional benefits of exoskeleton gait training for individuals with MS.

## 1. Introduction

Multiple Sclerosis (MS) is a chronic neurological condition that affects the central nervous system (CNS), specifically the brain and the spinal cord. MS is characterized by an abnormal immune reaction that causes a process called demyelination. Through this process, the immune system attacks the myelin, which is involved in the proper transmission of nerve signals along the various components of the CNS, causing areas of myelin sheet loss or damage. MS affects a wide range of neurological functions, like sensory input, processing speed, cognition, memory, attention [1,2], vision [3,4], and motor. In people with MS, reduced muscle strength and abnormal muscle tone (spasticity) lead to difficulties in performing activities like walking, standing, stair climbing and sit-to-stand, decreasing movement efficiency and coordination [5,6,7]. Balance control problems are often one of the earliest symptoms of MS and are common to both individuals with significant impairments and those with minimal impairments [8]. They manifest as a reduced ability to maintain position, reduced and slow movements toward their limit of stability, and a delayed response to postural perturbations [9]. In quiet standing, individuals with MS exhibit significantly greater postural sway compared to healthy controls, particularly under challenging conditions such as eyes closed. Moreover, oscillations of the Center of Pressure (CoP) have been shown to increase in correlation with higher levels of disability, as indicated by high scores on the Expanded Disability Status Scale (EDSS) [10]. Numerous studies have primarily ascribed balance impairments in MS to sensory deficits, including numbness and altered proprioception [11,12,13]. The critical role of the proprioceptive input in balance control has been well established in numerous studies on healthy subjects, in which proprioceptive input was artificially perturbated. Such perturbations led to inaccurate estimation of body segments positioning, increased body sway, and impaired balance control [14,15,16,17,18,19,20,21,22,23,24,25,26,27,28,29,30]. As a result of spasticity [13] and demyelination [12], individuals with MS exhibit delayed and/or distorted proprioceptive feedback [31,32]. Therefore, these impairments may underline their reduced ability to accurately perceive body segment positions and to generate appropriate motor responses necessary for maintaining postural stability [32]. In individuals with MS, balance impairments are often associated with abnormal gait, resulting in slower gait velocity, reduced step length, cadence, and joint motion, as well as prolonged swing phase duration, increased step width, and longer double support time compared to healthy controls [11]. Gait velocity and step length significantly decrease with increasing disability levels while the percentage of double support time increases in individuals with MS with a higher disability score [33].

The significant impact of balance and gait impairments on the quality of life has underscored the importance of rehabilitation interventions in individuals with MS. Various rehabilitation approaches have been explored, ranging from traditional physical rehabilitation to newer methods like dual-task training, robot-assisted walking training, virtual reality, and game-based training. Physical therapy and exercise programs have demonstrated significant positive effects on gait performance, particularly on gait speed [34,35,36,37,38]. Similarly, newer approaches, such as exoskeleton training, have shown to reach promising results in walking rehabilitation [39,40,41,42,43,44,45], as well as in the enhancement of balance control [46,47,48]. For neurodegenerative diseases, a partial assistance exoskeleton is frequently used for gait training. This type of exoskeleton assists gait by interacting with the user and contributing to the intended movement. This requires a synergy between the user’s intent and the device’s control, so that an appropriate exoskeleton torque or motion may be defined accordingly [49,50]. The need for active involvement of the individual makes this exoskeleton particularly suitable for increasing muscle strength and for enhancing the proprioceptive system, ultimately facilitating the restoration of sensorimotor function [46,48,51]. Both active and passive movements are associated with proprioceptive processing and can serve as forms of proprioceptive training. However, compared to passive training, active proprioceptive training has shown greater effectiveness in promoting somatosensory and motor neurophysiological outcomes [52,53,54,55].

This study aims to preliminarily investigate the effects of a four-week gait training program using the ExoAtlet II exoskeleton (ExoAtlet, Esch-sur-Alzette, Luxembourg) on static balance control, evaluated by the Center of Pressure analysis [56,57], in a small group of minimally impaired remitted individuals with MS (Expanded Disability Status Scale (EDSS) ≤ 2.5). Additionally, it explores the impact of this training on the Time Up and Go (TUG) test performance. Particular focus is given to the time spent to complete sub-phases requiring fine vertical balance control and where lower limb muscle strength is a critical determinant of balance [58,59], namely: the sit-to-stand and stand-to-sit phase, as well as the linear walking phase [60,61]. Previous studies have shown positive effects of the ExoAtlet II exoskeleton training on the walking function of individuals with MS in remission or secondary progressive course MS [62,63]. We therefore chose this partial assistive exoskeleton, which has been shown to improve muscle strength and enhance proprioceptive sensory feedback [46,48,51], hypothesizing that, taking advantage from these processes, its use may promote a more a-specific functional improvement, restoring healthy behavior. Hence, we expect not only an improvement in gait, for which the subjects are trained, but also in balance and in other functional activities requiring fine muscle control. This would suggest a generalization of the training effects, likely mediated by enhanced neuromuscular activation and improved proprioceptive feedback. Ultimately, such gait exoskeleton training approaches could contribute to a more comprehensive functional recovery, increasing autonomy in daily activities and enhancing overall quality of life. These findings would have important implications for rehabilitation, emphasizing the value of interventions that actively engage the subject with MS to foster broader motor improvements. In this preliminary study, we selected individuals with minimal MS impairment (in remission phase and EDSS ≤ 2.5), considering that this category of individuals would better tolerate the intensity of gait exoskeleton training compared to those with higher EDSS scores, thereby reducing the risk of muscle fatigue, which can lead to increased spasticity and suboptimal gait training outcomes [64]. Nonetheless, in spite of the low EDSS, we expect their baseline behavior to differ, in terms of fine motor control and balance abilities, from that of healthy subjects. Therefore, based on previous studies showing that individuals with low EDSS may easily develop mechanisms of neuronal plasticity when driven by partial assistive exoskeleton training [65,66], we expect that the proposed treatment may be effective on our group of individuals with mild MS impairment, leading them to recover a performance closer to that of controls.

## 2. Materials and Methods

### 2.1. Participants

Five individuals diagnosed with Multiple Sclerosis and fifteen healthy subjects (HSs) were recruited for the study.

The inclusion criteria for individuals with MS included: a confirmed medical diagnosis of MS, ability to ambulate with a Hauser Ambulation Index of 3 or less (meaning that they could walk independently for 8 m in a maximum time of 20 s), and, according to the technical specifications of the ExoAtlet II exoskeleton, a weight under 100 kg and height between 160 and 190 cm [67], no other comorbidities (e.g., cardiac deficiency, COPD, or respiratory failure), trunk and/or lower limb pain scores lower than 3/10 on a self-reported Visual Analog Scale (with 0 corresponding to “no pain at all” and 10 indicating “the worst pain ever experienced”), and compliance to use robotic devices. Healthy subjects were excluded if they had any musculoskeletal, neurological, or orthopedic disorder, walking impairments, or previous surgery involving their lower extremities.

For the five individuals with MS (3 females and 2 males), mean (± SD) age, height, and weight were 48.8 ± 10.7 years, 168.4 ± 9.2 cm, and 61 ± 15.7 kg. Their EDSS score ranged between 2 and 2.5. Participants were recruited from the Laboratory of Adapted Motor Activity (LAMA) of the Department of Public Health, Experimental Medicine, and Forensic Science of the University of Pavia, within a collaboration agreement between the LAMA and the Italian Multiple Sclerosis Association (AISM). The gait exoskeleton training period for all MS participants took place between February and April 2023 and all MS participants were in the remission phase of the disease. In order to assess the existence of a baseline difference between our mildly impaired subjects with MS and healthy individuals and to verify the extent to which the exoskeleton training could lead to recovering healthy abilities, we considered a set of 15 matched control subjects (8 females and 7 males), with mean age of 50 ± 10.8 years, height of 169.5 ± 4.1 cm, and weight of 67.2 ± 12.9 kg. Control subjects were recruited from the staff members of the Department of Electrical, Computer, and Biomedical Engineering of the University of Pavia. The two groups were similar for age (independent samples Student’s *t*-test, *p* = 0.83), weight (*p* = 0.4), height (*p* = 0.4), and gender distribution (Chi-squared test, *p* = 1).

All participants provided written informed consent to participate in the study, which received ethical approval from the local Ethics Committee of Pavia (Protocol number: 0030363/23). All the research procedures were performed in accordance with the Declaration of Helsinki.

### 2.2. Robotic Exoskeleton and Gait Training Session

Individuals with MS underwent gait training with an exoskeleton (ExoAtlet II, Esch-sur-Alzette, Luxembourg). The ExoAtlet II (Figure 1) is a CE-marked device designed to improve, in a rehabilitation context, the gait function of subjects with neurological or muscular injuries, diseases, or disabilities, to rebuild muscles, and to restore joint motion. This device has a metallic structure that surrounds the user’s lower limbs and trunk. In the walking task it assures mechanical assistance to the subject guiding the hip and knee angular joints to follow predefined kinematic trajectory profiles. Four electric motors and mechanical actuators move the hip and the knee, providing two degrees of freedom each. Conversely, the foot is supported by a passive spring ankle joint. By using the exoskeleton, the subject can better activate his/her muscles and move properly the lower limbs, correcting the pathological gait and deviating from a person’s habitual walking pattern. In rehabilitation treatments, this operating modality may induce adaptive phenomena in motor abilities. The device can be regulated to fit the subject’s physical size. The total weight of ExoAtlet II is around 33 kg. The kinematic gait characteristics can be set taking into account the exoskeleton operating mode (e.g., standing up, sitting down, stepping in place, walking, and climbing upstairs and downstairs). In this gait experimental set-up, the step length was set to 40 cm, the step height to 15 cm, the step duration to 1.2 s and the pause between steps was set to 0 s. These values allow to keep a natural, albeit relatively slow, gait pattern for all participants.

Each subject with MS underwent two gait exoskeleton training sessions per week, each one of a 30 min duration. The gait training lasted 4 consecutive weeks. Subjects were not involved in any other physiotherapy treatment during the study; however, no other motor activity restrictions were imposed. During the training session the subjects were asked to walk along a 10 m linear walkway, without rest breaks, at a speed of 1.3 km/h imposed by the exoskeleton. This is the maximum speed settable for the ExoAtlet II exoskeleton [67]. This velocity was low but safe and comfortable for the subjects wearing the exoskeleton, and it was not restrictive to obtaining training positive results in the walking function of the individuals with MS [62,64]. The experimenter walked behind the subject holding the “control handles” on the back of the exoskeleton. The control handles allowed the experimenter to start and stop the motor tasks when needed. Two rotational platforms for subject mobilization (Disco Duo, Chinesport S.p.A., Udine, Italy) were placed at the extremities of the walkway to facilitate the walking direction change and the rotation. When the subject reached one of the two rotational platforms the experimenter interrupted the exoskeleton walking and asked the subject to stand still while being passively rotated by means of the platform. Before starting with the gait training, the anthropometric measurements of each individual with MS were collected in order to personalize the length of the exoskeleton segments to the participant’s body. After that, the experimenter securely strapped the subject to the powered lower-limb exoskeleton to start with a 10 min familiarization gait session. The latter allowed the individuals with MS to learn how to shift their body weight from one foot to the other while following the exoskeleton movements and without resisting them. All participants completed all scheduled training sessions except for the subject MS5 due to work-related organizational problems.

### 2.3. Static Balance Test: Experimental Set-Up and Data Analysis

For individuals with MS, static balance evaluation was performed a week before (pre-Exo) and a week after (post-Exo) the exoskeleton training. HSs achieved only one evaluation session. Balance evaluation and the TUG test were performed in a randomized order across subjects. For balance evaluation, subjects were asked to stand on a baropodometric platform (FreeMed™ Platform, Sensor Medica, Roma, Italy) with arms straight and placed along their side and with their bare feet parallel and about 10 cm apart. Three trials with eyes closed (EC) and 3 trials with Eyes Open (EO), each lasting 30 s, were performed in a randomized order. The platform data recording began 5 s after a go signal of the operator in order to avoid any potential transitory adjustment at the beginning of the acquisition. During EO trials, subjects were asked to look straight ahead, avoiding head motion or rotation. The platform data were acquired at a sampling frequency of 25 Hz by dedicated software and then analyzed by a software developed in MATLAB (v. R2024a). The CoP displacement along the Antero–Posterior (AP) and the Medio–Lateral (ML) axes, after removing their respective mean values, were filtered with a high-pass Butterworth filter (4th order) with a cut-off frequency of 0.01 Hz. Then, the *Sway Area* was calculated as the area of the 95% confidence ellipse fitted to the statokinesigram; the *Standard Deviation* (*SD*) of the CoP displacement around its mean position along both the ML and the AP directions were calculated to evaluate the amplitude of the CoP displacement along each axis. The *Sway Density Curve* (*SDC*) was also considered. This time-versus-time curve was obtained following these two steps: first of all, for each time instant (t) the number of the statokinesigram consecutive samples that fall within a circle with a radius of 1 mm and centered in the current CoP(t) position was computed; secondly, each resulting number of samples was divided by the sampling frequency in order to obtain for each time instant the time spent by the CoP inside the moving circle [68]. In order to identify the peaks of the SDC, corresponding to the moments in which the subject was more stable (*Stay Time*), the curve was filtered with a low pass 4th order Butterworth filter with a cutoff frequency of 1 Hz [57]. Finally, for each subject and trial, the mean *Stay Time* value was calculated. Moreover, a cluster analysis, using the *dbscan* function of MATLAB, was also performed on the CoP statokinesigram. *Dbscan* clusters the CoP samples based on a threshold for a neighborhood search radius (1 mm) and a threshold for the minimum number of neighbor points (25 points, corresponding to 1 s) (Figure 2a). This 1 s treshold was set based on the works of Collins and De Luca [69], demonstrating, that during quiet standing, a critical time interval of about 1 s is the time necessary to switch from an open-loop to a closed-loop control of balance that exploits the afferent information from the visual, vestibular, proprioceptive, and somatosensory systems. Therefore, setting a treshold of 25 neighbor samples (1s), allowed to consider as a cluster only the time intervals in which sensory information gives an important contribution to body stabilization. CoP samples that did not meet the requirements imposed with the two thresholds were classified as outliers. Through this cluster analysis, regions of stability (CoP samples belonging to a cluster) and instability (CoP samples classified as outliers, red dots in Figure 2a) were identified for each trial. The classification of the CoP samples into clusters was reported on the SDC (Figure 2b), and the time interval in which CoP samples were classified as outliers was considered as a period of instability (paled-colored rectangles). The duration of each instability interval was calculated, and the identified instability periods were summed to obtain the *Total Instability Duration* for each trial.

Summarizing, the following CoP parameters were considered: *Sway Area*, *SD* along AP and ML directions, mean *Stay Time*, and *Total Instability Duration*. For each subject and for each parameter, the mean value (± SD) of three trials was computed. The mean and SD across subjects were also calculated for both the HS and individuals with MS (pre- and post-Exo) groups. The MS5 data were discarded from the mean and SD computation of the group of individuals with MS because this subject did not complete all the exoskeleton training sessions.

### 2.4. Dynamic Test: Experimental Set-Up and Data Analysis

The dynamic task evaluation was also performed pre- and post-Exo training for the individuals with MS. HSs achieved only one evaluation session. Subjects were asked to perform the TUG test wearing an inertial sensor (G-Sensor^®^, BTS Bioengineering, Milan, Italy) attached, using an adjustable elastic belt, to their back at about the L1 vertebra level. An arm-less chair was used. Subjects were asked to perform the TUG test as fast as possible, without running [70]. Subjects performed three TUG test trials. The inertial data were acquired and processed by a dedicated software (BTS G-Studio v. 3.5.27.0, BTS Bioengineering, Milan, Italy) that calculated the *Total Duration* of the test and the time spent to complete: the *sit-to-stand* phase, the *linear walking* phase (3 m forward walking + 3 m back to the starting point after turning around the cone), and the *stand-to-sit* phase. The different phases of the TUG test were expressed as a percent of the TUG *Total Duration*. For each subject and for each time phase duration the mean value (±SD) of the three trials was computed. Then the mean and SD across subjects were calculated for both HSs and pre- and post-Exo. In addition, for the TUG test, the MS5 data were not considered for the mean and SD computation across individuals with MS.

### 2.5. Statistical Analysis

Assumptions for parametric statistics were met for all variables of interest, as assessed by the Kolmogorov–Smirnov and Levene’s tests. In order to compare the balance performance between individuals with MS pre-Exo and HSs and between individuals with MS post-Exo and HSs a 3 (group: HSs, pre-Exo, and post-Exo) × 2 (visual condition: EO and EC) ANOVA was used to compare *Sway Area*, *SD* along AP and ML directions, *Stay Time*, and *Total Instability Duration*, separately. Moreover, a one-way ANOVA (group: HSs, pre-Exo and post-Exo) was used to compare each TUG variable (*Total Duration*, and the *sit-to-stand*, *stand-to-sit*, and *linear walking* phases). To evaluate the effect of the exoskeleton training in the group of individuals with MS a 2 (visual condition) × 2 (repetition: pre- and post-Exo) paired ANOVA was used to compare: *Sway Area*, *SD* along AP and ML directions, *Stay Time*, and *Total Instability Duration*, separately. A paired sample Student’s *t*-test was performed separately for the *Total Duration*, and *sit-to-stand*, *stand-to-sit*, and *linear walking* phase durations to assess the difference between pre- and post-Exo training in the individuals with MS. The durations of *sit-to-stand*, *stand-to-sit*, and the *linear walking*, expressed as a percentage of the TUG *Total Duration*, were compared after arcsine transformation to achieve a normal distribution of the data. Partial eta squared (η^2^_p_) was reported as a measure of effect size, with 0.01, 0.06, and 0.14 considered as small, medium, and large effect sizes, respectively [71]. For all ANOVAs, post-hoc analyses were performed using a Bonferroni test. The Cohen’s d effect size (with 0.2, 0.5, and 0.8 considered as small, medium, and large effect sizes, respectively) was also reported [72]. A *p*-value < 0.05 was considered statistically significant. Statistical analyses were performed using JASP (v.0.18.1, JASP Team 2023).

## 3. Results

### 3.1. Static Balance Test

Figure 3 shows the mean CoP position on the base of support, calculated separately for the subjects with MS and HSs, under both EO (Panel (a)) and EC (Panel (b)) conditions. During the pre-Exo evaluation (filled light green (EO) and blue (EC) circles), under both visual conditions, the mean CoP position in the group of subjects with MS was very close to that of the HS group (filled red circles, Panels (a,b)) and just a little further ahead under EC condition (Panel (b)). After the exoskeleton training the mean CoP position of the subjects with MS group (filled dark green (EO) and blue (EC) circles) were superimposable to that of HS group under both EO and EC conditions.

A clear example of the sway behavior of the individuals with MS before pre- and post-Exo training is shown in Figure 4, where the statokinesigrams under EO (Panel (a)) and EC (Panel (b)) conditions are exemplified for one representative subject with MS (MS1). For comparison, the statokinesigram of one representative healthy subject is shown under both EO (Panel (c)) and EC (Panel (d)) conditions. In this Figure, to help the visual comparison between the pre- and post-Exo training data and between these latter data and that of the healthy subject, the instantaneous CoP position was plotted with respect to the mean CoP position of each trial. As shown in the Figure, individual MS1 exhibits greater postural sway compared to the healthy subject in both the AP and ML directions and under both visual conditions (compare Panel (a) with (c) and Panel (b) with (d)) in the pre-Exo EO, and in both pre- and post-Exo EC. Furthermore, after the exoskeleton training the area covered by the CoP decreased in both visual conditions, indicating improved postural control.

Figure 5 summarizes the results concerning the balance parameters in the time domain. In panels (a,b,d,e,g,h), the mean values of each individual with MS pre- and post-Exo training are depicted separately for the EO and EC conditions. In each panel the red line corresponds to the mean parameter value across the HSs, while the dotted red lines indicate the mean ± SD. Panels (a,b) display the results of the mean *Sway Area*. Under the EO condition, subjects MS1 and MS2 demonstrated a reduction of the mean *Sway Area* post-Exo (of about 65% and 78% with respect to the pre-Exo value, respectively), reaching a mean value like that of the HSs. No changes between pre- and post-Exo were evident for subjects MS3 and MS4, while subject MS5, who had not completed the training, showed an increase in the mean *Sway Area* between pre- and post-Exo. Under EC condition, only subject MS1 improved their balance control after the training (i.e., a reduction of about 65% in the mean *Sway Area* value). The other individuals with MS showed comparable (MS3 and MS4) or increasing (MS2 and MS5) mean values between pre- and post-Exo. Panel (c) shows the mean *Sway Area* calculated across individuals with MS (subject MS5 excluded), for the pre- and post-Exo conditions separately, and across HSs. The two visual conditions are represented separately. *Sway Area* differed significantly between the EO and EC conditions (F(1,20) = 13.04, *p* < 0.01, η^2^_p_ = 0.39), no interaction between vision and group was found (F(2,20) = 1.37, *p* = 0.28, η^2^_p_ = 0.12). A marginally significant effect was observed in the comparison of the groups of subjects (F(2,20) = 3.34, *p* = 0.05, η^2^_p_ = 0.25). Under the EO condition, the *Sway Area* of the pre-Exo condition of the individuals with MS was higher than that of the HSs (post-hoc, *p* < 0.05, d = −0.49). Under the EC condition, this difference became even more pronounced, even if it was not statistically significant, probably due to the high variability of the data of the individuals with MS (see SD of light blue bar of Figure 5c) (post-hoc, *p* = 0.66, d = −1.68). After the training, the *Sway Area* of the individuals with MS decreased towards values comparable to that of the HSs. This was true for both the EO and EC conditions (post-hoc, *p* = 1, d = −0.08 and *p* = 1, d = −0.65, respectively). In summary, before initiating exoskeleton training, MS individuals demonstrated greater postural sway compared to HSs under both the EO and EC conditions. Following the intervention, a marked reduction in sway towards HS levels was observed across both conditions, making the results of the two groups comparable. Although this noticeable reduction between the pre- and post-Exo conditions (Figure 5c), when comparing these (one-way repeated measures ANOVA), the difference did not reach statistically significant values for either visual condition (F(1,3) = 3.26, *p* = 0.46, η^2^_p_ = 0.2).

Figure 5d,e show the mean values of the CoP oscillations (mean *SD*) along the AP axis of each subject with MS pre- and post-Exo training under EO and EC, respectively. In each panel the red line corresponds to the mean CoP oscillations value of HSs, while the dotted red lines indicate the mean ± SD. The histogram of panel (f) shows the mean values calculated across subjects with MS (subject MS5 excluded) and HSs, grouped by visual condition. Under EO condition (Figure 5d), subjects MS1, MS2 and MS3 demonstrated a reduction of the CoP oscillations in the AP direction after the exoskeleton training (of about 63%, 57% and 40% with respect to the pre-Exo value, respectively), reaching a mean value like that of HSs. No changes between pre- and post-Exo were evident for subjects MS4 and MS5. Under EC condition (Figure 5e), a reduction of the CoP oscillation was evident for subject MS1 (57%) whereas the other subjects with MS showed an increase in the CoP oscillation in the AP direction. In the ML direction under EO condition (Figure 5g), only subjects MS2 and MS4 showed a reduction of the CoP oscillation after the exoskeleton training (of about 59% and 27% with respect to the pre-Exo value, respectively). Subjects MS1 and MS3 did not show any changes in the ML CoP oscillation between pre- and post-Exo, while MS5 slightly increased the oscillation amplitude after the training. Under EC condition (Figure 5h), a reduction of the CoP oscillation was evident for subjects MS1 and MS4 (33% and 25%, respectively). For the other individuals with MS no changes occurred, except for MS5 whose oscillation increased. Panels (f) (AP direction) and (i) (ML direction) of Figure 5 show the mean CoP oscillations calculated across subjects with MS under pre- and post-Exo training and across HSs. The mean results of both visual conditions are depicted separately. There was a significant effect of visual condition in the AP direction (F(1,20) = 17.51, *p* < 0.001, η^2^_p_ = 0.47), no interaction was found between vision and group (F(2,20) = 1.02, *p* = 0.38, η^2^_p_ = 0.09). For HSs, the CoP oscillations under EC condition were greater than those under EO condition. For *SD* CoP in AP direction, a marginally non-significant effect was observed when comparing pre- and post-Exo data of individuals with MS with that of HSs and (F(2,20) = 3.25, *p* = 0.06, η^2^_p_ = 0.24). In EO condition, pre-Exo *SD* in AP direction was higher than that of the HSs (post-hoc, *p* < 0.05, d = −0.79). In EC condition, this difference was also evident, but it did not reach significance (post-hoc, *p* = 1, d = −1.46). Post-Exo results where comparable to those of HSs. This was true for both EO and EC conditions (post-hoc, *p* = 1, d = −0.10 and *p* = 1, d = −0.9, respectively). In summary, before initiating exoskeleton training, individuals with MS demonstrated greater postural sway along the AP direction compared to HSs in both EO and EC conditions. Following the intervention, a marked reduction towards HS levels was observed across both conditions making the results of the two groups comparable. Although this appreciable reduction between pre- and post-Exo condition (Figure 5f) when comparing these results in both visual conditions, the difference did not reach statistical significance (F(1,3) = 0.58, *p* = 0.5, η^2^_p_ = 0.16).

Vision did not affect *SD* CoP ML values (F(1,20) = 2.63, *p* = 0.12, η^2^_p_ = 0.12). No significant effects were found among groups (F(2,20) = 2.55, *p* = 0.10, η^2^_p_ = 0.20) and in the interaction between vision and groups ((F(2,20) = 0.53, *p* = 0.59, η^2^_p_ = 0.05). In EO condition post-Exo value was smaller than that of pre-Exo condition. However, comparing pre-and post-Exo conditions no significant effect of the exoskeleton training was observed (F(1,3) = 6.33, *p* = 0.09, η^2^_p_ = 0.68).

The mean results of the *Stay Time* parameter are shown in Figure 6a–c. With EO (Panel (a)), the mean *Stay Time* increased for subjects MS1 (74%) and MS2 (103%) after the exoskeleton training, approaching the mean *Stay Time* value calculated for the HSs. Subjects MS3 and MS4 exhibited no change in the mean *Stay Time* parameter between the experimental sessions performed before and after the exoskeleton training, while subject MS5 showed again a worsening (i.e., shorter *Stay Time* after training). Under EC condition (Panel (b)), only the subject MS1 had an increase of the mean *Stay Time* parameter value after the exoskeleton training (176%). The other individuals with MS presented no changes in this parameter between pre- and post-Exo mean results, except for subject MS5, who displayed a decrease in mean *Stay Time* value. As shown in Panel (c), with EO the *Stay Time* is higher than that of the EC condition (F(1,20) = 34.06, *p* < 0.001, η^2^_p_ = 0.63). For the HS group the *Stay Time* EO mean value (14.8 ± 3.5 s) was significantly higher than that of the EC condition (9.4 ± 3.6 s; post-hoc, *p* < 0.001, d = 1.26). In addition, for the individuals with MS the *Stay Time* was longer under EO than EC condition, but this difference was significant only in the post-Exo condition (13.8 ± 4.9 s and 7.7 ± 4.8 s, respectively; post-hoc, *p* < 0.05, d = 1.43) as well as for HS subjects (post-hoc, *p* < 0.001, d = 1.26). No significant differences were found when comparing pre- and post-Exo data with that of HSs (F(2,20) = 0.86, *p* = 0.44, η^2^_p_ = 0.08). As for the other balance parameters no significant differences emerged when comparing pre- and post-Exo data (F(1,3) = 0.93, *p* = 0.41, η^2^_p_ = 0.24). However, an effect of the exoskeleton training on the mean *Stay Time* was observable under EO condition. With EO, the mean *Stay Time* showed a slight increase (post-Exo > pre-Exo, post-hoc, *p* = 0.94, d = −0.61), indicating a tendency for the CoP to remain in a center of stability for more time, approaching a value similar to the mean value calculated across HSs (EO condition, HS vs. post-Exo, *p* = 1, d = 0.23).

The mean value of the *Total Instability Duration* calculated for each individual with MS is reported in panel (d,e) of Figure 6. Under the EO condition (Panel (d)), a reduction of the mean *Total Instability Duration* occurred when comparing pre- and post-Exo values for the MS1 (66%), MS2 (68%), MS4 (43%), and MS5 (45%) subjects. Subject MS3, whose value was already similar to that of the HSs, showed no changes following exoskeleton training. With EC (Panel (e)), when comparing pre- and post-Exo values, the mean *Total Instability Duration* decreased for subjects MS1 (61%) and MS4 (28%), approaching the mean value of *Total Instability Duration* calculated across HSs. For subjects MS2 and MS5 the mean *Total Instability Duration* increased, while for subject MS3 the mean value showed no substantial changes after the training period. In panel (f) of Figure 6, the mean *Total Instability Durations* across the four individuals with MS are depicted for the two visual conditions and the pre- and post-Exo conditions. The mean *Total Instability Duration* was significantly higher under the EC than EO condition (F(1,20) = 11.14, *p* < 0.01, η^2^_p_ = 0.36). Only in the HSs was this difference significant (post-hoc, *p* < 0.05, d = −1.11). There was no difference in the mean *Total Instability Duration* between subject groups (F(2,20) = 0.30, *p* = 0.74, η^2^_p_ = 0.03). When comparing pre-Exo results with those of the post-Exo condition, no differences were found (F(1,3) = 0.27, *p* = 0.64, η^2^_p_ = 0.08). However, an effect of the exoskeleton training on the mean *Total Instability Duration* parameter was visible under the EO condition (post-hoc, *p* = 0.43, d = 0.52). With EC, exoskeleton training did not seem to have any effect on the *Total Instability Duration* parameter (post-hoc, *p* = 1, d = 0.09).

### 3.2. TUG Test

The mean values of the TUG test parameters computed across subjects are reported in Table 1 for both HSs and individuals with MS pre- and post-Exo. Individuals with MS spent more time completing the TUG test with respect to HSs (F(2,20) = 6.88, *p* < 0.01, η^2^_p_ = 0.41; see *Total Duration* parameter results). This was true for the pre-Exo condition (post-hoc, *p* < 0.05, d = −1.72), and although the difference was reduced post-Exo training, a difference from the HSs still persisted (post-hoc, *p* < 0.05, d = −1.51). When comparing the results of the pre- and post-Exo conditions, the difference in MS TUG *Total Duration* did not reach significance (paired *t*-test, *p* = 0.68, d = 0.23). Analyzing the different phases of the TUG test, there was a difference between subject groups in the *stand-to-sit* phase (F(2,20) = 5.18, *p* < 0.05, η^2^_p_ = 0.34), but not in the *sit-to-stand* phase (F(2,20) = 0.28, *p* = 0.76, η^2^_p_ = 0.027) or *linear walking* phase (F(2,20) = 0.58, *p* = 0.57, η^2^_p_ = 0.05). Under the pre-Exo condition, individuals with MS required a longer time to complete the *stand-to-sit* phase with respect to the HSs (post-hoc, *p* < 0.05, d = −1.81). A similar trend was observed for the *linear walking* phase, although the difference did not reach statistical significance (*p* = 1, d = −0.47). Post-Exo, the duration of the *stand-to-sit* phase was notably reduced, approaching the time recorded for HSs (post-hoc, HS vs. MS post-Exo, *p* = 1, d = −0.45). Again, for the all TUG sub-phases’ durations, no differences were observed when comparing pre-Exo versus post-Exo conditions (paired *t*-test: *stand-to-sit*: *p* = 0.26, d = 0.69; *sit-to-stand: p* = 0.58, d = −0.31; and *linear walking*: *p* = 0.99, d = 5 × 10^−4^).

## 4. Discussion

In this preliminary investigation, we explored the feasibility of exoskeleton-based gait training and its effects on balance control, under both static and dynamic conditions, in a small group of minimally impaired individuals with MS (in remission phase and EDSS ≤ 2.5). Our results show the existence of a clear difference in static balance abilities relative to healthy controls, although statistically significant only for the condition in which visual inputs were available, probably due to the high variability of the data from the EC condition. This behavioral difference between the two tested populations also manifested during the TUG test’s execution, with individuals with MS requiring extended completion times for both the overall test and the stand-to-sit phase (see pre-Exo condition versus HSs).

Previous studies have demonstrated the effectiveness of an ExoAtlet II training on gait and cognitive function [62,63]. The aim of the present work was to evaluate the efficacy of exoskeleton-assisted gait training on balance, specifically assessing post-treatment recovery of static and dynamic balance performance compared to healthy controls. Therefore, the objective was not to compare the treatment’s effectiveness against conventional physiotherapy, but rather to determine the extent to which this treatment could be successful in restoring healthy behavior in our group of mildly impaired individuals with MS. The training consisted of walking for thirty minutes while wearing the exoskeleton, twice a week for four consecutive weeks. All five individuals with MS were able to accomplish the task required during each training session and all tolerated the exoskeleton well. All individuals with MS, except one (subject MS5), completed the four weeks of training. Subject MS5 was forced to suspend the training after two weeks for working engagements. A static posturographic test and TUG test were performed before and after the training session to assess static and dynamic balance performance. The same tests were administered to a group of matched HSs to establish a set of reference data for comparison with the performance of the individuals with MS. Due to the small sample size of the MS group, the results before and after the training session of this group were compared for each subject separately, then all the subjects’ data were pooled together for each experimental condition and statistically analyzed.

The results referring to HSs confirm those found in the literature about the static balance test under EO and EC conditions. Balance control is more precise when the CNS takes advantage of visual information compared to when it must rely solely on other sensory inputs, such as proprioception and the vestibular system [17,68,73,74,75,76,77]. Our preliminary results from individuals with MS suggest that gait exoskeleton training has a positive effect on balance control, promoting recovery towards healthy behavior under both static and dynamic conditions. Overall, a clear trend of improvement was observed across the parameters considered. Notably, in the baseline (pre-Exo) EO condition, data were visibly higher than the HS values, although without always reaching statistical significance. Following training, the results for individuals with MS showed a trend towards the HS values, and the previously observed significant differences between the two groups disappeared. A similar trend occurred under the EC condition; however, improvements in this condition were less evident, and the data were more variable.

### 4.1. General Comments About Balance Behavior of Individuals with MS Under Static and Dynamic Conditions: Comparison Between HSs and Pre-Training Results

To explore potential postural changes during quiet standing in individuals with MS that may be attributed to the disease, the mean CoP position of MS subjects was compared to that of HSs, under both EO and EC conditions. For the purposes of this analysis, we assumed that the CoP position on the base of support corresponds to the ground projection of the body’s center of mass, which represents the postural alignment maintained by the subject to remain in a steady state of equilibrium [78,79]. Under the EO condition, no noticeable differences were observed upon visual inspection, suggesting that, despite the presence of the disease, no postural alignment changes had developed during static upright stance. Conversely, under the EC condition, individuals with MS maintained a posture that was slightly tilted forward compared to that of HSs. However, when the CoP oscillations were investigated, individuals with MS performed worse than HSs, under both the EO and EC conditions. *Sway Area*, *SD* CoP AP and ML, and *Total Instability Duration* mean values were all greater than those recorded for the HSs. Concurrently, the mean *Stay Time* results were reduced in the subjects with MS. These differences were consistently observed across the analyzed variables under both the EC and EO conditions. However, only EO *Sway Area* and the *SD* CoP AP results showed statistically significant differences between the two groups. This can probably be explained by the greater sensitivity of these two parameters as indicators of balance control capability. The former reflects the capacity to maintain the center of gravity’s position within the base of support, necessitating major (large *Sway Area*) or minimal (small *Sway Area*) corrections. The latter, is consistent with the fact that in static conditions the most pronounced evidence of instability is typically observed in the AP direction rather than the ML [9,80]. Under the EC condition, individuals with MS exhibited a large variability, which may preclude statistically significant findings in the comparison between the two groups. Conversely, the enhanced data consistency and repeatability observed under the EO condition can be explained by the visual system’s compensatory role for diminished or absent proprioceptive input, a common characteristic in individuals with MS [17,75,76]. This enables them to sustain a moderate level of balance, even if below that observed in HSs. This finding appears even more reasonable when referring to subjects with a mild disease level (EDSS < 2.5) [11].

Our findings show a trend in line with previous results showing differences in postural control in individuals with MS in comparison with matched HSs [11,12,80,81,82], with marked disparities under challenging conditions, such as when the eyes are closed and visual information is denied [12,80,81,83,84]. To maintain postural control, the CNS relies on and continuously processes visual, vestibular, proprioceptive, and plantar cutaneous afferent information. It has been suggested that the information carried by individual sensory channels is combined and a weight is assigned to the various input sources depending upon the current functional state of a particular sensory system, the postural task itself, and the context in which task is being performed. This sensory integration process transforms input signals into higher-level neural signals that represent physical variables such as body or limb orientation and motion in space or with respect to the environment [74,85,86,87]. All these variables are fundamental to correctly estimate the position of the body center of mass and consequently to efficiently control balance. Unavailable or unreliable sensory input determines a conflict of sensory information that challenges postural control [88] as observed in individuals with MS, who displayed greater sway and prolonged intervals of instability, likely due to compromised proprioceptive feedback consequence of the muscle spasticity and the demyelination effect of the disease [11,80,82,89]. These differences become more evident under the EC condition, confirming that in individuals with impaired proprioception, there is a common tendency to a greater reliance on visual and vestibular cues to maintain balance. In the absence of these cues, the sensory integration process becomes more challenging, often resulting in suboptimal postural control [12,80,81,83,84]. Moreover, individuals with MS exhibited grater sway than HSs in both AP and ML directions under both EO and EC conditions, with a clear tendency to sway more in the AP direction. This phenomenon was again more pronounced under the EC condition. These findings are consistent with those previously reported by several authors [9,80].

The inefficiency in the balance control of individuals with MS affects not only the performance in static posture but also that of the TUG test. Specifically, we observed a statistically significant increase of the *Total Duration* (6.8 ± 0.6 s and 8 ± 0.8 s for HSs and subjects with MS, respectively), which is in agreement with the EDSS score of the subjects with MS [90,91], a mild increase of the *linear walking* phase duration (32.9 ± 3.6% and 33.9 ± 5.8% for HSs and individuals with MS, respectively), and a statistically significant increase of the time needed to complete the *stand-to-sit* task (16.5 ± 2.4% and 20.9 ± 2.3% for HSs and subjects with MS, respectively). As previously discussed in the literature, the increase of the TUG *Total Duration* in individuals with MS observed in our investigation, as well as in many other studies [60,91], could be ascribed to the neuroaxonal injury of these subjects [92], which is associated with a poor hip adductor strength [93] and a poor knee muscle strength [90,94]. These latter factors also affect the *linear walking* phase, leading to a slight increase of its duration compared to that observed in HSs. During walking, particularly in phases where the individual is standing on one foot and shifting the body forward, motor synergies involving the hip and knee muscles are crucial for maintaining balance and ensuring smooth step progression. When these muscles are weakened or impaired, a disruption of natural gait patterns can be observed [11,33,95]. Moreover, analyzing the duration of the other TUG phases we observed remarkable difficulties in individuals with MS when performing the *stand-to-sit* phase, resulting in a lengthening of this phase compared to what is observed in HSs. This is probably due to the highly demanding nature of this task, which requires optimal coordination, balance, adequate mobility, and sufficient strength and muscle power [60]. All these requirements are markedly compromised in individuals with MS, leading to a slower execution of this vertical movement.

### 4.2. Efficacy of Exoskeleton Gait Training in Restoring Healthy Balance Behavior in Static and Dynamic Conditions: Comparison Between HSs and Post-Training Results

Overall, we investigated whether wearing an exoskeleton during gait rehabilitation training induced changes in body posture when subjects stood in steady upright stance position. No postural changes were observed in individuals with MS under the EO condition after the training period, suggesting that the exoskeleton’s weight and the postural alignment imposed by its rigidity did not negatively affect the body schema of standing posture, which appears to be well consolidated in this population. Conversely, under the EC condition a slight improvement in postural alignment was observed, with the average CoP position of individuals with MS shifting closer to that of the HSs.

Several studies have assessed the effects of predominantly gait-focused rehabilitation treatment on balance control and gait performance in individuals with MS, reporting encouraging results [46,48,96,97,98]. Most of them introduced training using a robotic exoskeleton [48], end-effector robot [46,96], or treadmill with body weight support [95] for enhancing gait rehabilitation. Published evidence on the feasibility and preliminary efficacy of robotic exoskeleton-assisted exercise rehabilitation for improving mobility and balance in individuals with significant MS-related disability highlights the notion that actively engaging patients in gait training may enhance the likelihood of functional adaptation through activity-dependent neuroplasticity [48,99]. Assisted walking represents a highly complex motor behavior that depends on the rapid processing and integration of multisensory inputs. While robotic exoskeletons are designed to correct pathological gait patterns, their effective use requires substantial motor learning and cognitive engagement. Successful gait performance using assistive exoskeletons involves the integration of proprioceptive, locomotor, postural, visuospatial, and cardiorespiratory sensory information. This information must be processed in real time to generate accurate and adaptive motor responses. Such behavior activates a wide range of sensory receptors and pathways, engaging numerous interconnected neurophysiological processes regulated by the CNS. Evidence from Androwis and colleagues [43] demonstrates that exoskeleton-assisted gait rehabilitation can significantly enhance both motor and cognitive functions in individuals with MS. Based on the principles of neural plasticity, it is plausible to hypothesize that training-induced neural adaptations may generalize not only to the cognitive processes but also to other functions sharing overlapping neural circuits, such as balance control or other daily motor tasks. Consequently, we predicted that exoskeleton-assisted gait rehabilitation would produce broader functional benefits beyond locomotion alone. Our findings demonstrate that after gait training the *Sway Area, the SD* CoP AP and ML, and the *Total Instability Duration* were reduced, approaching values similar to those observed in the group of matched controls, under both the EO and EC conditions. This enhancement was also supported by an increase in the *Stay Time* parameter. No statistically significant differences were observed between the two groups for any parameter under either the EO or EC condition. Notably, under the EC condition, despite an observable post-treatment improvement in both performance values and data variability, the offset from healthy values remains greater than under the EO condition. Moreover, not all subjects improved their performance after the training: some individuals showed worse CoP control post-Exo than pre-Exo. A possible explanation could be the difficulty of the CNS in integrating sensory information without visual information [62]. Under this condition, the subject’s oscillatory behavior is less patterned if compared to that with eyes open. This is observed in healthy subjects and becomes more evident in subjects with neurological disorders. Even with a potential enhancement of proprioceptive input, ascribed to the training process, this information still fails to be processed correctly when combined with vestibular and somatosensory data. As a result, no clearly visible improvements in balance control were noticeable for all individuals with MS in the absence of vision. Consequently, our intervention using a partial assistance exoskeleton designed to enhance lower limb muscle strength and improve proprioceptive input [46,48,96,100,101] resulted in improved balance control, towards healthy performance, with this improvement being more pronounced under the EO condition. This outcome was observed despite the training’s primary focus on gait, thereby supporting the hypothesis of a transfer of training effect.

The results from the TUG test and its sub-phases further corroborate our hypothesis, as these phases represent the execution of key activities of daily living, such as walking, sitting down, and standing up. After the training, a remarkable reductions in the *Total Duration* and the *stand-to-sit* duration phase of the TUG test were observed, as well as a mild reduction in the *linear walking* phase duration. Prior to training, the *Total Duration* and the *stand-to-sit* phase were both statistically significantly greater than those observed in HSs. Post-intervention, these disparities diminished, resulting in the data becoming statistically comparable to the HS data. These promising findings suggest that four weeks of training are sufficient to elicit the mechanisms for which the exoskeleton was engineered (brain function plasticity, muscle strength, enhancement of proprioceptive inputs [46,48,96,100,101]) that are functional not only for gait improvement but also for balance control. Previously, Tavazzi and colleagues [100] found that after four-week gait rehabilitation treatment, individuals with MS showed a reduction in the magnitude of diffuse activation related to the motor task and functional connectivity in the precentral and postcentral gyrus bilaterally, thus concluding that the short-term beneficial effect of motor rehabilitation on walking performance in subjects with MS was accompanied by a functional reorganization of the brain’s sensory-motor network.

Finally, while the efficacy of exoskeleton gait training in restoring healthy balance behavior was clearly documented, no statistically significant differences were found for all static balance and TUG test parameters when pre- and post-exoskeleton conditions of individuals with MS were compared (a paired two-way ANOVA and a paired *t*-test for balance and TUG test parameter results, respectively). Indeed, the limited number of subjects (four) makes the results of these statistical tests less reliable and unrepresentative. This is evidenced by the computed statistical power values, which were below 20%, thereby substantially restricting the ability to detect meaningful changes.

## 5. Limits and Tips for Future Research

Despite the encouraging results this study presents several limits. First, the small sample size and related low statistical power, especially when the pre- and post-Exo conditions were compared. A larger sample size would have allowed including a randomized design and further comparisons of different experimental conditions, with a group of subjects not undergoing any rehabilitation and one treated with a traditional rehabilitation approach. Second, the lack of stratification by EDSS levels among individuals with MS. This further limits the generalizability of the results to the broader population with MS. Third, the lack of a longer follow-up period. This precludes verification of the consolidation of the mechanism underlying balance improvement. Fourth, the lack of an experimental set-up to objectively quantify the enhancement of lower limb muscle strength and proprioceptive input after exoskeleton training. Lastly, the available exoskeleton characteristics did not allow, probably for subjects’ safety issues, for a faster, more natural walking speed, and therefore prevented the possibility to explore different values of such parameters. Future research should collect data from a larger number of individuals with MS, incorporating stratification based on pathology level. Moreover, new experimental designs capable of exploring different facets of the efficacy of exoskeleton gait training on balance should be defined and implemented. Finally, these novel experimental setups will integrate electromyography data to assess improvements in muscle activity post-training and to monitor muscle fatigue resulting from exoskeleton sessions. Furthermore, the enhancement of proprioceptive input in static balance control could be investigated using foam surfaces or lower limb muscle vibration during static balance testing.

## 6. Conclusions

The current pilot study provides preliminary evidence in support for exoskeleton-assisted gait rehabilitation as a potential approach towards improving functional mobility and balance functions in mildly impaired individuals with MS, restoring ability similar to that of HSs. We speculated that these improvements are the result of an adaptive plasticity behavior based on the strengthening of muscles and enhancement of proprioceptive feedback. If confirmed from further investigations involving a larger cohort of individuals with MS, these results may have significant implications for improving motor function that extends beyond the clinical setting to a multitude of daily life activities and needs. Such functional gains are crucial, as they contribute directly to increased autonomy and participation in everyday activities, ultimately leading to a better quality of life.

## Figures and Tables

**Figure 1 bioengineering-12-00826-f001:**
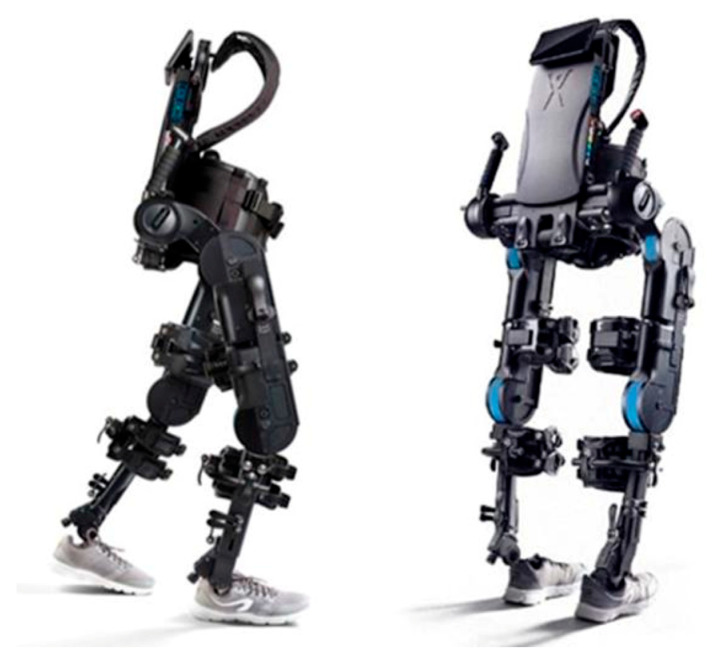
The ExoAtlet II exoskeleton: side and rear views.

**Figure 2 bioengineering-12-00826-f002:**
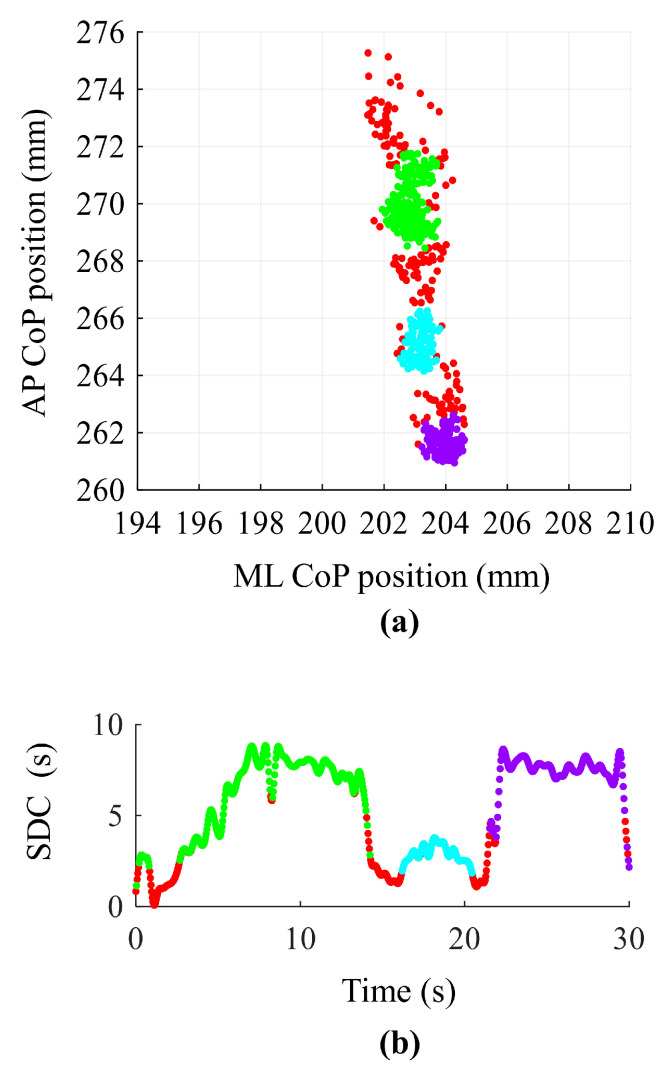
Cluster analysis and the Sway Density Curve (SDC). (**a**) The statokinesigram (Antero–Posterior (AP) position versus Medio–Lateral (ML) position of the Centre of Pressure (CoP)) of one representative subject with MS under the Eyes Open (EO) condition before the Exoskeleton training (pre-Exo). Referred to this trial, the *dbscan* function identified, on the statokinesigram, three clusters (green, light blue, and violet dots) considered as the areas of stability. The red dots are the outliers identified by the *dbscan* algorithm. (**b**) The SDC curve of the same trial depicted in Panel (**a**). The CoP samples classification obtained by the cluster analysis is superimposed to the SDC curve in order to compute the period of instability (paled coloured rectangles) of this trial.

**Figure 3 bioengineering-12-00826-f003:**
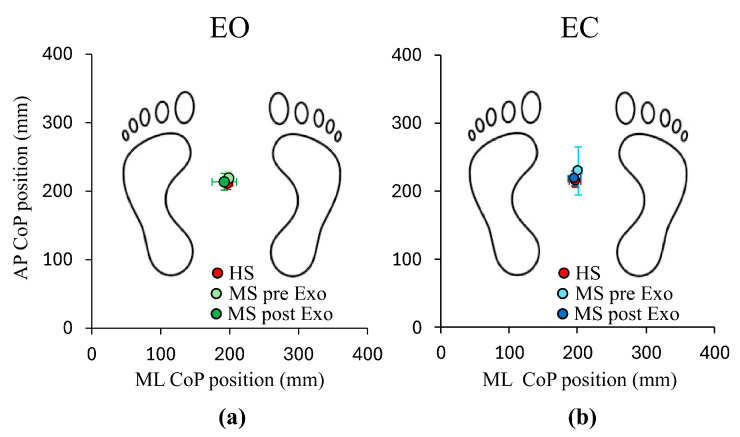
Mean (± SD) Center of Pressure (CoP) position on the base of support (Antero–Posterior (AP) axis versus Medio–Lateral (ML) axis) under Eyes Open (EO) (**a**) and Eyes Close (EC) (**b**) conditions. The filled light green and blue circles correspond to the mean data of pre-Exo evaluation (pre-Exo) EO and EC values of individuals with Multiple Sclerosis (MS), respectively, whereas the filled dark green and blue circles refer to the data of post-Exo evaluation (post-Exo). The red circles correspond to the mean CoP position calculated across the Healthy Subjects (HSs) under both EO and EC conditions.

**Figure 4 bioengineering-12-00826-f004:**
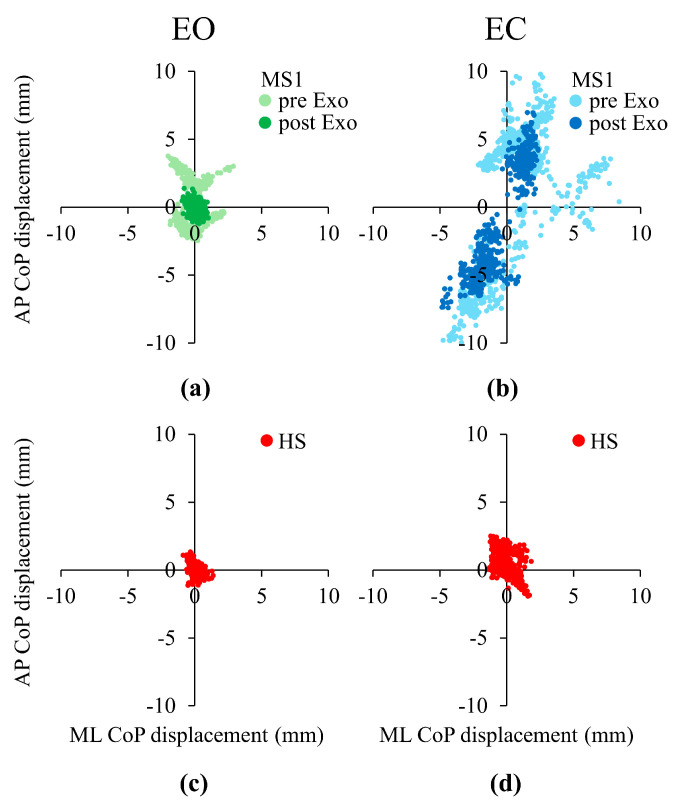
Center of Pressure (CoP) statokinesigram (Antero–Posterior (AP) axis versus Medio–Lateral (ML) axis). The CoP statokinesigram of subject MS1 is reported under Eyes Open (EO) (**a**) and Eyes Closed (EC) (**b**) conditions for one trial of both pre-Exoskeleton evaluation (pre-Exo, light green circles in (**a**) and light blue circles in (**b**)) and post-Exoskeleton evaluation (post-Exo, dark green circles in (**a**) and dark blue circles in (**b**)) evaluations. For comparison, in panels (**c**,**d**) a trial of one representative healthy subject (HS) is reported for the EO (**c**) and EC (**d**) conditions.

**Figure 5 bioengineering-12-00826-f005:**
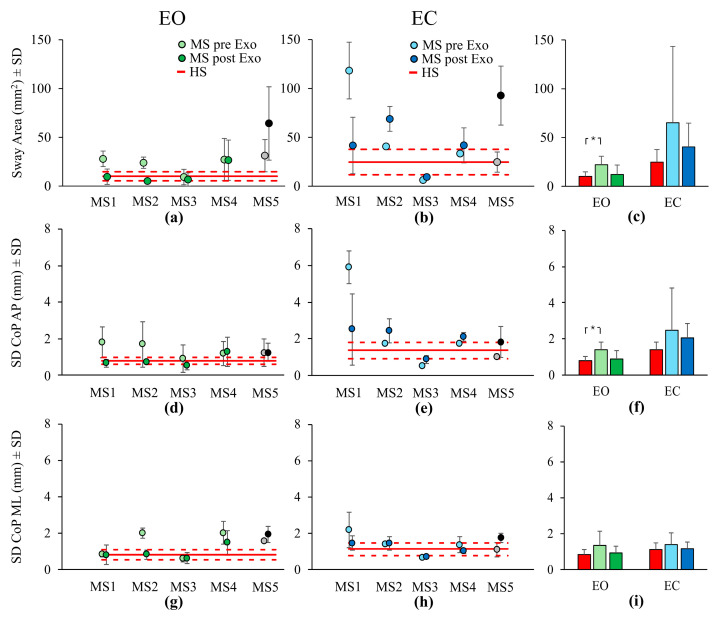
In panels (**a**,**b**) (mean *Sway Area* data), (**d**,**e**) (mean *SD* CoP along AP axis), and (**g**,**h**) (mean *SD* CoP along ML axis) the light green and light blue filled circles represent the mean pre-exoskeleton (pre-Exo) evaluation data for individuals with MS, while the dark green and dark blue filled circles represent the post-exoskeleton (post-Exo) evaluation data. Individual mean data for each subject with MS is shown separately. The grey and black filled circles correspond to the pre-Exo and post-Exo evaluations data of subject MS5, who was excluded from the group average calculations as their training sessions were not completed. The red line indicates the mean value across healthy subjects (HSs), and the red dotted lines represent the mean ± Standard Deviation (SD). In Panels (**c**) (mean *Sway Area*), (**f**) (mean *SD* CoP along the AP axis), and (**i**) (mean *SD* CoP along the ML axis), the red bars indicate the mean across HSs, the green bars (light and dark) represent the mean across individuals with MS under the Eyes Open (EO) condition (pre- and post-Exo, respectively), and the blue bars (light and dark) represent the mean across individuals with MS under the Eyes Closed (EC) condition (pre- and post-Exo, respectively). Asterisks indicate significant differences (*, *p* < 0.05). EO, Eyes Open; EC, Eyes Closed; CoP, Center of Pressure; AP, Antero–Posterior; ML, Medio–Lateral; HS, healthy subject; MS, Multiple Sclerosis; Exo, Exoskeleton; and SD, Standard Deviation.

**Figure 6 bioengineering-12-00826-f006:**
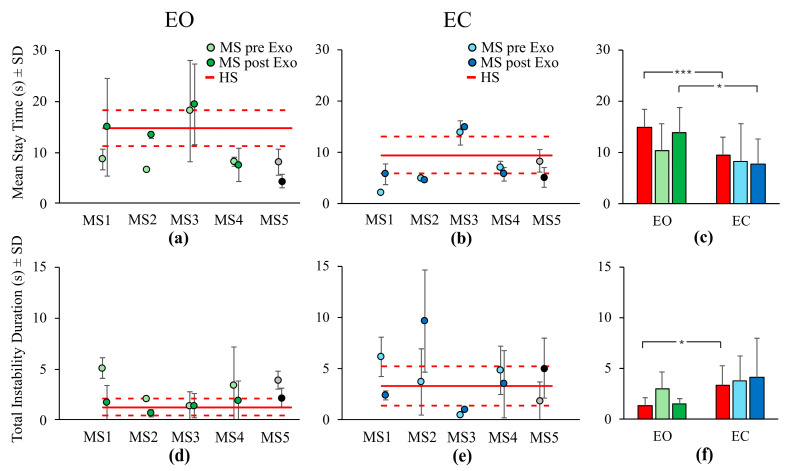
In panels (**a**,**b**) (mean *Stay Time*), (**d**,**e**) (mean *Total Instability Duration*), the light green and light blue filled circles represent the mean pre-exoskeleton (pre-Exo) evaluation data for individuals with MS, while the dark green and dark blue filled circles represent the post-exoskeleton (post-Exo) evaluation data. Individual mean data for each subject with MS is shown separately. The grey and black filled circles correspond to the pre-Exo and post-Exo data of the subject MS5, who was excluded from the group average calculations as their training sessions were not completed. The red line corresponds to the mean value across HSs and the red dotted lines represent the mean ± SD. In panels (**c**) (mean *Stay Time*) and (**f**) (mean *Total Instability Duration*), the red bars indicate the mean across HSs, the green bars (light and dark) represent the mean across individuals with MS under EO condition (pre-Exo and post-Exo, respectively) and the blue bars (light and dark) represent the mean across individuals with MS under EC condition (pre-Exo and post-Exo, respectively). Asterisks indicate significant differences (*, *p* < 0.05; ***, *p* < 0.001). EO, Eyes Open; EC, Eyes Closed; HSs, healthy subjects; MS, Multiple Sclerosis; Exo, Exoskeleton; SD, Standard Deviation.

**Table 1 bioengineering-12-00826-t001:** Timed Up and Go test. The mean time taken by healthy subjects (HSs) and subjects with Multiple Sclerosis (MS) pre- and post-Exo training to complete the TUG test is reported. The duration of the different phases of the TUG test (*sit-to-stand*, *stand-to-sit*, and *linear walking*) are computed as percent of the total duration of the test. The mean time (seconds) of each phase is also reported in parentheses.

	HS	MS Pre-Exo	MS Post-Exo
*Total Duration* (s)	6.8 ± 0.6	8.0 ± 0.8	7.7 ± 0.6
*sit-to-stand* (%Total Duration)	16.8 ± 2.4 (1.12 ± 0.1 s)	16.3 ± 1.8 (1.29 ± 0.05 s)	16.8 ± 3.6 (1.3 ± 0.3 s)
*stand-to-sit* (%Total Duration)	16.5 ± 2.4 (1.13 ± 0.2 s)	20.9 ± 2.3 (1.66 ± 0.1 s)	17.9 ± 3 (1.4 ± 0.3 s)
*linear walking* (%Total Duration)	32.9 ± 3.6 (2.2 ± 0.3 s)	33.9 ± 5.8 (2.54 ± 0.7 s)	31.1 ± 2.9 (2.25 ± 0.2 s)

## Data Availability

Data will be made available upon request.

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
