# Peer review of "Four-Week Exoskeleton Gait Training on Balance and Mobility in Minimally Impaired Individuals with Multiple Sclerosis: A Pilot Study"

_bioengineering, 2025, doi:10.3390/bioengineering12080826_

Round 1
Reviewer 1 Report
Comments and Suggestions for Authors
This manuscript focuses on an interesting topic that can have a great impact on the quality of life of patients with multiple sclerosis . However, it presents some aspects that should be subject to further revision.
This reviewer misses explicit information on what is new about this work compared to similar work in the area. On the other hand, there are aspects related to the participants in the study that should be explained. I will mention the details later in the report, in the appropriate section.
Abstract
This section mentions that the participants were "minimaly impaired". It should be specified with some parameter, since that is not a clinical criterion.
It is explained that center of pressures were measured and the TUG test was performed. It should be explained what variable or what parameter those tests are intended to determine.
The conclusions are not the concrete ones derived from the study, but general and too ambitious for the results obtained.
INTRODUCTION
It is comprehensive, neatly presented and easy to read.
Perhaps the prominence given to walking speed should be minimized. Gait speed is a very important parameter in the quality of life of people, especially those with multiple sclerosis, but this manuscript has not studied them.
Considering that parameters related to static and dynamic equilibrium are measured, these aspects should be discussed, their difference, their importance, and how to evaluate them.
The text that appears between lines 87 and 104 should go at the end of the introduction, since it is the objective.
The hypothesis should (if included) appear before the objective.
The text that appears between lines 105 and 111 should be in the methodology section.
METHODOLOGY
Information is missing on where and how participants were recruited (word of mouth, association, hospital, rehabilitation center...)
The dates on which the study was conducted should be included. If all participants did the training with the exoskeleton at the same time, or different times of the year.
Multiple sclerosis is a chronic disease, but it presents multiple manifestations and types depending on the progression of the disease. It also presents the possibility of relapse periods and seasonal variations (hence the importance of the commentary in the previous paragraph). All this must be taken into account and explained accordingly in the introduction, methodology and discussion sections.
Among the inclusion criteria, the 100 kg weight and height of the participants are probably conditioned by the exo specifications. If so, it should be mentioned.
Where were the healthy participants recruited? Were they only age-matched? And by sex, BMI?
A section on statistical analysis and data processing is missing. The authors make it clear that this is a pilot program and that because of this, because of the reduced n, some improve and others get worse and that is why they have not done any statistical analysis. Perhaps this should be explained in methodology, and discussed in a limitations section within the discussion. All this in relation to the types of MS and the possible seasonal variations of the disease, possibility of outbreaks...
Lines 125-128: this information should be included in the results section as: “descriptive characteristics of the participants”.
RESULTS
In general, all acronyms should be developed in table headings and figure captions, even if they have been previously described in the body of the text.
Another aspect that should be clarified is that "measuring the center of pressure" and performing "the TUG" are not the same as the parameters underlying these tests, which are in fact the variables that are analyzed.
Although the N is small, the authors should do some statistical analysis, to show at least a trend in case the results are not significant (both to see if there are differences between healthy individuals and MS, and before and after the intervention).
Considering that the manuscript is presented as a pilot project, somewhere in the discussion it should be indicated which would be the future option to demonstrate its hypothesis.
Author Response
This reviewer misses explicit information on what is new about this work compared to similar work in the area. On the other hand, there are aspects related to the participants in the study that should be explained. I will mention the details later in the report, in the appropriate section.
The Introduction section has been extensively revised, with a particular focus on the rationale for selecting individuals with MS in the context of exoskeleton use. We also emphasize the novelty of our study concerning the current literature on ExoAtlet II training for MS individuals. To date, no studies have investigated the outcomes of this training on balance control; rather, all existing research highlights its positive effects on gait, cognitive functions, and hand motor skills
ABSTRACT
This section mentions that the participants were "minimaly impaired". It should be specified with some parameter, since that is not a clinical criterion.
This information is now added reporting the score of the Expanded Disability Status Scale of the patients (line 19).
It is explained that center of pressures were measured and the TUG test was performed. It should be explained what variable or what parameter those tests are intended to determine.
All the details about CoP quantification and TUG test parameters were now reported (from line 21 to line 23)
The conclusions are not the concrete ones derived from the study, but general and too ambitious for the results obtained.
The conclusions section has been modified avoiding general speculations and limiting the discussion to the presented results (from line 24 to line 28)
NTRODUCTION
Considering that parameters related to static and dynamic equilibrium are measured, these aspects should be discussed, their difference, their importance, and how to evaluate them.
In the last part of the Introduction we highlight the different aspects of balance control that can be investigated analyzing both the CoP data acquired during static posturographic test and the TUG test data. Moreover, the parameters classically used for the data analysis of these two tests are also introduced. To this end, new studies have been cited (from line 87 to line 93).
The text that appears between lines 87 and 104 should go at the end of the introduction, since it is the objective.
This is done.
The text that appears between lines 105 and 111 should be in the methodology section.
This text has been removed from the Introduction.
METHODOLOGY
Information is missing on where and how participants were recruited (word of mouth, association, hospital, rehabilitation center...)
All these information were added in this revised version of the manuscript (from line 129 to line 132).
The dates on which the study was conducted should be included. If all participants did the training with the exoskeleton at the same time, or different times of the year. Multiple sclerosis is a chronic disease, but it presents multiple manifestations and types depending on the progression of the disease. It also presents the possibility of relapse periods and seasonal variations (hence the importance of the commentary in the previous paragraph). All this must be taken into account and explained accordingly in the introduction, methodology and discussion sections.
Thank you for pointing this fundamental issue. For all patients, the entire rehabilitation program was scheduled in order to avoid performing any training session in the summer season. All the patients were in the remission phase. These information are now specified (from line 132 to line 134)
Among the inclusion criteria, the 100 kg weight and height of the participants are probably conditioned by the exo specifications. If so, it should be mentioned.
We mentioned this point referring to the guidelines of the ExoAtlet II (line 121). This new reference has been added.
Where were the healthy participants recruited? Were they only age-matched? And by sex, BMI?
All these information were included in the Participants section. Statistical tests were performed providing evidence of the correlation between the two groups considered in the study. (from line 136 to line 139)
A section on statistical analysis and data processing is missing. The authors make it clear that this is a pilot program and that because of this, because of the reduced n, some improve and others get worse and that is why they have not done any statistical analysis. Perhaps this should be explained in methodology, and discussed in a limitations section within the discussion. All this in relation to the types of MS and the possible seasonal variations of the disease, possibility of outbreaks...
A new section intitled Statistical analysis has been included. In this section all the statistical tests used to analyze the results are described in detail (from line 269 to line 283).
Lines 125-128: this information should be included in the results section as: “descriptive characteristics of the participants”.
The statistical results pertaining to the matching between the two groups are not considered the focus of the study and, therefore, have been reported in the description of the healthy subjects within the Material and Methods section (from line 136 to 139).
RESULTS
In general, all acronyms should be developed in table headings and figure captions, even if they have been previously described in the body of the text.
This is done in this revised version.
Another aspect that should be clarified is that "measuring the center of pressure" and performing "the TUG" are not the same as the parameters underlying these tests, which are in fact the variables that are analyzed.
The different meanings of the CoP parameters and those of the TUG test are described in the Material and Methods section. (from line 202 to line 235 and from 250 to 255).
The results of the variables referring to the CoP were described separately from those of TUG test. Moreover, the data are depicted in different ways: the CoP parameters were shown with graphs whereas the results of the TUG test were summarized in one table.
Although the N is small, the authors should do some statistical analysis, to show at least a trend in case the results are not significant (both to see if there are differences between healthy individuals and MS, and before and after the intervention).
All the statistical results are now described in the Results section.
Considering that the manuscript is presented as a pilot project, somewhere in the discussion it should be indicated which would be the future option to demonstrate its hypothesis.
In this regard a new section entitled Limits and tips for future research, has been included (from line 712 to line 724)

Reviewer 2 Report
Comments and Suggestions for Authors
This pilot study examined the effects of a 4-week exoskeleton gait training program on static and dynamic balance among individuals with multiple sclerosis. Five individuals with multiple sclerosis and 15 age-matched healthy individuals were enrolled in this study. One week before and after the 4-week training, posturography and functional mobility were assessed for the multiple sclerosis group. The control group received no training, and the assessment was only done once. The comparisons in the outcome measures indicated that robotic-assisted training could improve static balance under the open-eyed condition and functional mobility, while the improvement in static balance with eyes closed was subtle. Although the topic of this manuscript falls within the scope of the journal, a few major concerns have dampened its quality and impact. Please see specific comments below.
- Consider using people first language throughout the manuscript.
- Lines 81-83. Please provide the references supporting this statement.
- A review of the literature regarding the application of the exoskeleton suit of interest (ExoAtlet) in people with multiple sclerosis should be conducted. This could strengthen the rationale and scientific premise for conducting this study.
- Line 150. Is “1.2 s” the step or strike duration?
- Line 157. The gait speed during the training is listed as 1.3 km/h, which is equivalent to 0.361 m/s. This speed is much slower than the regular walking speed for individuals with multiple sclerosis with an EDSS ranging between 2 and 2.5. With this under-challenging walking speed, it is unclear how the exoskeleton suit would improve the motor performance of standing and walking.
- 1a shows that the CoP displacement in the ML direction appears to be unreasonably small (less than 4 mm). Is this correct?
- A statistical analysis section is missing in the Methods section.
- The unit of measurement for the CoP is inconsistent throughout the manuscript (mm vs. cm).
- 4. Some individuals with MS showed worse COP performance after the training than pre-training for the EC condition. This observation should be discussed.
- The manuscript also lacks a report of statistical results. Without the results, it is unfounded or less convincing to conclude whether the training made improvements or not.
- Line 400. There is an extra period.
- The Discussion section does not seem to deliberate the potential mechanisms by which the exoskeletal suit improves static and dynamic balance, which is unclear. The improvement in static balance and functional mobility could result in enhanced sensation, increased strength, and reduced fear of falling, among other benefits. Since there are no measurements of these potential factors, it is hard to attribute the improvement in the outcome measures to these factors.
- The healthy control seems unnecessary, as the improvement was considered within the training group.
- The manuscript lacks a limitation section. The potential limitations include a small and uneven sample size, the lack of an active control group, and a non-randomized controlled design, among others. These limitations should be acknowledged.
Author Response
- Consider using people first language throughout the manuscript.
This is done in this new revised version.
- Lines 81-83. Please provide the references supporting this statement.
Three new references have been added (line 80).
- A review of the literature regarding the application of the exoskeleton suit of interest (ExoAtlet) in people with multiple sclerosis should be conducted. This could strengthen the rationale and scientific premise for conducting this study.
Two new references regarding the effects of an ExoAtlet II exoskeleton training in a group of remitted MS individuals have been added (from line 93 to 95).
- Line 150. Is “1.2 s” the step or strike duration?
It is a step duration.
- Line 157. The gait speed during the training is listed as 1.3 km/h, which is equivalent to 0.361 m/s. This speed is much slower than the regular walking speed for individuals with multiple sclerosis with an EDSS ranging between 2 and 2.5. With this under-challenging walking speed, it is unclear how the exoskeleton suit would improve the motor performance of standing and walking.
In this revised version, we addressed this issue by specifying that this is the maximum configurable velocity for the ExoAtlet II exoskeleton. Furthermore, we discuss this setting in relation to both the patients and the required task (from line 170 to 175).
- 1a shows that the CoP displacement in the ML direction appears to be unreasonably small (less than 4 mm). Is this correct?
We have verified and confirmed that the data depicted in Fig. 1 are correct. These results align with previous studies analyzing CoP data from patients suffering from multiple sclerosis (Kanekar, N.; Lee, Y.-J.; Aruin, A.S. Frequency analysis approach to study balance control in individuals with multiple sclerosis. J. Neurosci. Methods 2014, 222, 91–96. doi: 10.1016/j.jneumeth.2013.10.020).
- A statistical analysis section is missing in the Methods section.
In this revised version we added one paragraph entitled Statistical Analysis. In this paragraph all statistical tests performed to compare the data are detailed (from line 269 to 283).
- The unit of measurement for the CoP is inconsistent throughout the manuscript (mm vs. cm).
Now all the measurement units are correctly reported, and they are all consistent throughout the manuscript.
- Some individuals with MS showed worse COP performance after the training than pre-training for the EC condition. This observation should be discussed.
With vision denied the CNS experienced difficulty in integrating sensory information which manifests as an increased variability of CoP data results in static upright posture. In this condition the subject's oscillatory behavior is less patterned if compared with that with eyes open. This is observed in healthy subjects and becomes more evident in patients with neurological disorders. This is now discussed at the end of the Discussion session (from line 696 to 706).
- The manuscript also lacks a report of statistical results. Without the results, it is unfounded or less convincing to conclude whether the training made improvements or not.
All the statistical results are now described in the Results section.
- Line 400. There is an extra period.
In this version the extra period is removed.
- The Discussion section does not seem to deliberate the potential mechanisms by which the exoskeletal suit improves static and dynamic balance, which is unclear. The improvement in static balance and functional mobility could result in enhanced sensation, increased strength, and reduced fear of falling, among other benefits. Since there are no measurements of these potential factors, it is hard to attribute the improvement in the outcome measures to these factors.
The discussion has been reorganized and extensively revised, aiming to avoid drawing conclusions not supported by the results as much as possible.
- The healthy control seems unnecessary, as the improvement was considered within the training group.
In this revised version, we explain the reasoning behind this choice in the Discussion section (from line.528 to line 534).
- The manuscript lacks a limitation section. The potential limitations include a small and uneven sample size, the lack of an active control group, and a non-randomized controlled design, among others. These limitations should be acknowledged.
A new section entitled Limits and tips for future research, has now been included (from line 712 to line 724).

Reviewer 3 Report
Comments and Suggestions for Authors
This is an interesting study that aimed to investigate the benefits of a four-week exoskeleton gait training program on balance and mobility in individuals with MS. Despite its quality and alignment with the journal’s scope, I believe the manuscript requires substantial revision. Please review my suggestions carefully, as the paper would benefit from a more comprehensive and rigorous approach.
Main Weakness: The authors did not conduct any statistical tests to assess the effects of the exoskeleton training on MS participants. I strongly recommend the inclusion of statistical analyses, as potential benefits can only be meaningfully interpreted through robust testing. Specifically, I suggest performing a two-way repeated measures analysis of variance (ANOVA), comparing the effects across group (MS vs. controls), time (pre- vs. post-intervention), and their interaction. Reporting effect sizes is also important, even in pilot studies. Ideally, the study should include an additional MS group undergoing conventional exercise training (e.g., physiotherapy) to strengthen the comparison.
Title: As the study involves two groups, consider including the term “case-control” before “pilot study” to more accurately reflect the design.
Abstract: Clarify the definition of “minimally impaired MS patients” by including the Expanded Disability Status Scale (EDSS) scores. The results should be described in greater detail. Again, I strongly recommend performing a two-way repeated measures ANOVA to compare outcomes across groups and time points, including interaction effects. Reporting effect sizes is important for interpreting the practical significance of findings.
Introduction: The introduction is well written—congratulations to the authors. However, I suggest breaking up long paragraphs to improve readability. Additionally, the authors should explore bioengineering concepts further and explain how the exoskeleton may benefit MS patients who are functionally near-normal (described as “minimally impaired”). If participants are only mildly affected, the rationale for using an exoskeleton needs to be clearly justified. I recommend adding 2–3 paragraphs focusing on biomechanical reasoning to support the potential benefits for this population (“minimally impaired” MS participants).
Methods: Provide more detail on participant selection. Explain why only five individuals with MS were included compared to 15 healthy controls. Including an image of the exoskeleton would help readers better understand the intervention. As emphasized above, statistical analysis is crucial to properly interpret the findings. I cannot recommend acceptance of this manuscript without appropriate statistical testing—even for a pilot study.
Results: Describing results without statistical testing raises the possibility that the findings are influenced by uncontrolled or random factors. While the narrative is clear, the lack of pre-post comparisons between MS and control groups severely limits the interpretability of the results.
Discussion: The absence of robust statistical analysis makes it impossible to determine whether the findings are truly attributable to the exoskeleton or to other factors. Furthermore, applying an exoskeleton to minimally impaired participants must be carefully justified. Rather than using healthy controls, would it not be more informative to include a second MS group undergoing conventional physiotherapy?
Conclusion: Appropriate.
References: Avoid excessive self-citation. The current manuscript includes four references from the authors’ own group. While this may not seem excessive in the context of 92 total references, self-citation should be limited. I suggest reducing this to no more than two references from the authors’ group.
Author Response
Main Weakness: The authors did not conduct any statistical tests to assess the effects of the exoskeleton training on MS participants. I strongly recommend the inclusion of statistical analyses, as potential benefits can only be meaningfully interpreted through robust testing. Specifically, I suggest performing a two-way repeated measures analysis of variance (ANOVA), comparing the effects across group (MS vs. controls), time (pre- vs. post-intervention), and their interaction. Reporting effect sizes is also important, even in pilot studies. Ideally, the study should include an additional MS group undergoing conventional exercise training (e.g., physiotherapy) to strengthen the comparison.
In this revised version, the statistical analysis of the results has been done. The statistical design is outlined in a dedicated section entitled Statistical Analysis (from line 269 to283). The statistical results are inserted and discussed throughout the Results and the Discussion sections.
Title: As the study involves two groups, consider including the term “case-control” before “pilot study” to more accurately reflect the design.
We appreciate your suggestion. However, we've opted to retain the previously proposed title to give emphasis solely to individuals with MS. As explained in the discussion, our hypothesis was that exoskeleton treatment would enhance static and dynamic balance performance in MS patients, enabling them to achieve balance levels approximating those of healthy subjects. Data from healthy subjects were collected exclusively to establish these reference values.
Abstract: Clarify the definition of “minimally impaired MS patients” by including the Expanded Disability Status Scale (EDSS) scores. The results should be described in greater detail. Again, I strongly recommend performing a two-way repeated measures ANOVA to compare outcomes across groups and time points, including interaction effects. Reporting effect sizes is important for interpreting the practical significance of findings.
The Abstract has been comprehensively revised, incorporating all previously missing details. However, we chose to omit the statistical results due to the template's word count constraints (200 words).
Introduction: The introduction is well written—congratulations to the authors. However, I suggest breaking up long paragraphs to improve readability. Additionally, the authors should explore bioengineering concepts further and explain how the exoskeleton may benefit MS patients who are functionally near-normal (described as “minimally impaired”). If participants are only mildly affected, the rationale for using an exoskeleton needs to be clearly justified. I recommend adding 2–3 paragraphs focusing on biomechanical reasoning to support the potential benefits for this population (“minimally impaired” MS participants).
The Introduction has been comprehensively reviewed and enriched with all your helpful suggestions. Among the clarifications incorporated, the justification for the selection of mildly affected individuals with MS has now been delineated (from line 107 to 113).
Methods: Provide more detail on participant selection. Explain why only five individuals with MS were included compared to 15 healthy controls. Including an image of the exoskeleton would help readers better understand the intervention. As emphasized above, statistical analysis is crucial to properly interpret the findings. I cannot recommend acceptance of this manuscript without appropriate statistical testing—even for a pilot study.
All these suggestions were considered, and the details have been integrated into this revised version. The statistical analysis has been performed, and the results are now described and discussed.
Results: Describing results without statistical testing raises the possibility that the findings are influenced by uncontrolled or random factors. While the narrative is clear, the lack of pre-post comparisons between MS and control groups severely limits the interpretability of the results.
All the statistical results are now described in the Results section.
Discussion: The absence of robust statistical analysis makes it impossible to determine whether the findings are truly attributable to the exoskeleton or to other factors. Furthermore, applying an exoskeleton to minimally impaired participants must be carefully justified. Rather than using healthy controls, would it not be more informative to include a second MS group undergoing conventional physiotherapy?
All these suggestions and concerns have been addressed throughout this revised version. More specifically, the Introduction section (from line 107 to line 113) details the rationale for selecting and recruiting minimally impaired MS individuals. Furthermore, in the first part of the Discussion section, we clarify why we did not compare the Exo Atlet II training results with those of conventional physiotherapy (lines 528-534).
Conclusion: Appropriate.
References: Avoid excessive self-citation. The current manuscript includes four references from the authors’ own group. While this may not seem excessive in the context of 92 total references, self-citation should be limited. I suggest reducing this to no more than two references from the authors’ group.
One self-citation was removed.

Round 2
Reviewer 1 Report
Comments and Suggestions for Authors
The authors have done a great job in revising the manuscript.
Note: For my part, only the year in line 137 remains to be added.
Author Response
Dear Reviewer
the year has been added in this version of the manuscript (line 142).
We sincerely appreciate the time and effort you invested for improving the quality of our manuscript.
Best regards
Reviewer 2 Report
Comments and Suggestions for Authors
The authors have addressed most of the previous concerns. However, a few of them still need attention. Several extra comments arose during the reading of the revised manuscript. Please see specific notes below.
- (Original comment #1). The authors claimed that they have revised the manuscript to use the people-first language in the revised manuscript. However, this is not the case at all.
- (Original comment #13). The authors stated that they had “explain[ed] the reasoning behind this choice [of using healthy controls] in the Discussion section.” Yet, nothing on the specified lines mentioned the rationale for using the healthy control group in this study. As stated before, the inclusion of this control group offers little benefits to the overall purpose of this study. However, it considerably and unnecessarily complicates the study design, statistical model, and results interpretation.
- (Original comment #14). One of the major limitations – the non-randomized study design – is not mentioned in the newly added limitation section. Other missing limitations include the lack of a real control group (the current one did not undergo the training), the slow and constant peak walking speed of the exoskeletal device across participants, etc.
- The authors used unnecessarily excessive numbers of abbreviations, which severely impede the reading of this manuscript. Also, the way of using the abbreviations is not consistent. For example, some of the acronyms are repeatedly defined. The authors did not stick to the abbreviations after their definitions. Instead, they used the full names in some cases.
- Lines 99-101. This hypothesis is problematic as none of the mentioned possible mediators, including leg muscle activation, muscle strength, proprioception, were measured in this manuscript. Therefore, such hypothesis is foundationless.
- Lines 115-117. The rationale for this hypothesis is unclear. Additionally, nothing was done to test this hypothesis in this manuscript.
- It is unclear why five individuals with multiple sclerosis were enrolled. Was any power analysis conducted?
- Also unclear is why the body mass and height were restricted (lines 124-125).
- Another major concern is the primary statistical approach – ANOVA with repeated measures. The outcome measures were assessed only once for the healthy control group but twice for the group with multiple sclerosis. Please note that repeated measures ANOVA are not appropriate when one group has only one assessment and the other has two. Repeated measures ANOVA is designed for situations where the same individuals are measured under multiple conditions or at multiple time points. For this study, there are an unequal number of measurements across the groups, violating this core principle. The control group with one assessment has a single data point, while the other group with two assessments has two, making the data types of incompatible for a repeated measures design. Therefore, the statistical approach could be inappropriate and lead to invalid and unreliable results and findings.
- Given the large number of outcome measures, the risk of inflating the Type I error is high. The significance level should be adjusted to suppress such a risk, which would likely ensure the validity and robustness of the conclusions.
Author Response
Dear Reviewer
We are grateful for the opportunity to revise our manuscript, "Four-weeks exoskeleton gait training on balance and mobility in minimally impaired individuals with multiple sclerosis: a pilot study," once again. We sincerely appreciate the time and effort you invested in providing us with invaluable comments for improving its quality. We have thoroughly revised the manuscript in accordance with your suggestions. For clarity, all changes in the manuscript are marked in blue. Below, we provide a detailed, point-by-point response to the reviewer’s comments, explaining how each suggestion has been addressed in this revised version.
The authors have addressed most of the previous concerns. However, a few of them still need attention. Several extra comments arose during the reading of the revised manuscript. Please see specific notes below.
1. The authors claimed that they have revised the manuscript to use the people-first language in the revised manuscript. However, this is not the case at all.
We apologise if not all corrections were previously implemented. We have now replaced "MS patients" with "individuals with MS" or "subjects with MS" throughout the manuscript.
2. (Original comment #13). The authors stated that they had “explain[ed] the reasoning behind this choice [of using healthy controls] in the Discussion section.” Yet, nothing on the specified lines mentioned the rationale for using the healthy control group in this study. As stated before, the inclusion of this control group offers little benefits to the overall purpose of this study. However, it considerably and unnecessarily complicates the study design, statistical model, and results interpretation.
In this new version, we have included a sentence in the Materials and Methods section (lines 143-148) clarifying the rationale for including the healthy control group in the study. This aspect is then revisited in the discussion section (lines 520-533), where the text has been partially modified. While we recognise that integrating healthy control data adds complexity to the statistical design, we consider that the presence of a reference behaviour (healthy subjects) for comparing patients' baseline and post-treatment outcomes is of significant interest. This comparison will contribute to validating the hypothesis that exoskeleton treatment in individuals with mild EDSS, who nonetheless exhibit reduced performance compared with healthy controls, can facilitate the recovery of a behaviour that is closer to that of the healthy.
3. (Original comment #14). One of the major limitations – the non-randomized study design – is not mentioned in the newly added limitation section. Other missing limitations include the lack of a real control group (the current one did not undergo the training), the slow and constant peak walking speed of the exoskeletal device across participants, etc.
In this revised version, we have included all these suggestions about the study's limitations (lines 722-726 and 731-737).
4. The authors used unnecessarily excessive numbers of abbreviations, which severely impede the reading of this manuscript. Also, the way of using the abbreviations is not consistent. For example, some of the acronyms are repeatedly defined. The authors did not stick to the abbreviations after their definitions. Instead, they used the full names in some cases.
We have removed the three abbreviations pertaining the TUG sub-phases and the one referring to the static balance test parameter. Moreover, we verified the consistency of the abbreviation throughout the manuscript. The repetition of abbreviation definitions in figure captions and table headings was a request from Reviewer 1 and, therefore, these have been retained in this revised version as well.
5. Lines 99-101. This hypothesis is problematic as none of the mentioned possible mediators, including leg muscle activation, muscle strength, proprioception, were measured in this manuscript. Therefore, such hypothesis is foundationless.
We recognize the previous statement's lack of clarity and poor structure. In this revision, we clarified that our hypothesis was grounded in prior experimental evidences from other Authors, which led us to postulate the exoskeleton's effectiveness for a-specific rehabilitation program (lines 99 -103).
6. Lines 115-117. The rationale for this hypothesis is unclear. Additionally, nothing was done to test this hypothesis in this manuscript.
The sentence has undergone revision to improve its clarity (lines 115-121).
7. It is unclear why five individuals with multiple sclerosis were enrolled. Was any power analysis conducted?
The limited patient cohort is due to the fact that the collaboration among the University of Pavia, the ExoAtlet company, and the Lunex University, established under the SAMURAI project, was confined to a nine-month period. During this timeframe, the exoskeleton was transferred from Lunex University to the University of Pavia, enabling researchers to familiarize themselves with the device through preliminary testing before patient deployment. This process consumed considerable time, as did the iterative refinement of a definitive protocol that would be sustainable for individuals with MS without inducing fatigue; this task necessitated numerous preliminary experimental trials. Furthermore, despite the engagement of the Lama Laboratory of the University of Pavia with the MS association, recruiting subjects willing to participate in such an extensive rehabilitation protocol was challenging.
8. Also unclear is why the body mass and height were restricted (lines 124-125).
In this version, it has been explained (lines 128-129).
9. Another major concern is the primary statistical approach – ANOVA with repeated measures. The outcome measures were assessed only once for the healthy control group but twice for the group with multiple sclerosis. Please note that repeated measures ANOVA are not appropriate when one group has only one assessment and the other has two. Repeated measures ANOVA is designed for situations where the same individuals are measured under multiple conditions or at multiple time points. For this study, there are an unequal number of measurements across the groups, violating this core principle. The control group with one assessment has a single data point, while the other group with two assessments has two, making the data types of incompatible for a repeated measures design. Therefore, the statistical approach could be inappropriate and lead to invalid and unreliable results and findings.
We agree with your comment and thank you for raising this point. The previous statistical design, implemented based on a suggestion from Reviewer 3, presented the issue of data repetition for healthy subjects (HS) to facilitate paired data comparisons. In this revised version, the statistical design has been redefined. Specifically, for balance parameters a 3 (group: HS, pre-Exo and post-Exo) × 2 (visual condition: EO and EC) ANOVA was performed to compare pre-Exo and HS data, and post-Exo and HS data, considering both visual conditions. A 1-way ANOVA (group: HS, pre-Exo and post-Exo) was also conducted for TUG parameters. Furthermore, to compare the pre-Exo and post-Exo data within the group of subjects with MS, a 2 (visual condition: EO and EC) x 2 (repetition: pre- and post-Exo) paired ANOVA was utilized for balance parameters, and a paired samples Student’s t-test was applied for TUG parameters. The statistical analysis section has been revised accordingly (lines 280-293).
10. Given the large number of outcome measures, the risk of inflating the Type I error is high. The significance level should be adjusted to suppress such a risk, which would likely ensure the validity and robustness of the conclusions.
We have considered your suggestion; however, given the preliminary nature of the data and the primary goal of identifying trends to inform future larger-scale research, no corrections for multiple comparisons have been applied to limit the inflation of type I error. This decision has been supported by the guidelines of Kianifard & Islam (F. Kianifard & M.Z. Islam. A guide to the design and analysis of small clinical studies. Pharmaceutical Statistics, 10(4), 363-368, 2011) and Schochet (P.Z. Schochet. Technical Methods Report: Guidelines for Multiple Testing in Impact Evaluations. NCEE 2008-4018. National Center for Education Evaluation and Regional Assistance, 2008).

Reviewer 3 Report
Comments and Suggestions for Authors
The authors did an excellent job enhancing the quality of the study. All my suggestions were either incorporated or properly justified. I am satisfied with the current version of the text.
Author Response
Dear Reviewer
We sincerely appreciate the time and effort you invested for improving the quality of the our manuscript.
Best regards